# snoRNPs: Functions in Ribosome Biogenesis

**DOI:** 10.3390/biom10050783

**Published:** 2020-05-18

**Authors:** Sandeep Ojha, Sulochan Malla, Shawn M. Lyons

**Affiliations:** 1Department of Biochemistry, Boston University School of Medicine, Boston, MA 02115, USA; sojha@bu.edu (S.O.); smalla2012@my.fau.edu (S.M.); 2The Genome Science Institute, Boston University School of Medicine, Boston, MA 02115, USA

**Keywords:** snoRNA, snoRNP, ribosome biogenesis, pseudouridylation, 2′-*O*-methylation, RNA processing

## Abstract

Ribosomes are perhaps the most critical macromolecular machine as they are tasked with carrying out protein synthesis in cells. They are incredibly complex structures composed of protein components and heavily chemically modified RNAs. The task of assembling mature ribosomes from their component parts consumes a massive amount of energy and requires greater than 200 assembly factors. Among the most critical of these are small nucleolar ribonucleoproteins (snoRNPs). These are small RNAs complexed with diverse sets of proteins. As suggested by their name, they localize to the nucleolus, the site of ribosome biogenesis. There, they facilitate multiple roles in ribosomes biogenesis, such as pseudouridylation and 2′-*O*-methylation of ribosomal (r)RNA, guiding pre-rRNA processing, and acting as molecular chaperones. Here, we reviewed their activity in promoting the assembly of ribosomes in eukaryotes with regards to chemical modification and pre-rRNA processing.

## 1. Introduction

Chemical modifications of RNA are found throughout all domains of life. The MODOMICS database currently lists 172 experimentally confirmed RNA modifications [1], although recent data suggests that there could be other unclassified RNA modifications [2]. Recent advances have begun to shed light on the role of RNA modifications in messenger (m)RNA, yet the most heavily modified cellular RNAs are transfer (t)RNAs and ribosomal (r)RNAs. Combined, these two classes comprise more than 90% of all cellular RNA, so the relative importance of RNA modification on these molecules cannot be understated. Modifications of rRNA are widespread throughout all domains of life [3,4]. Ribosomes are incredibly complex macromolecular machines comprised of 80 proteinaceous components and four noncoding RNAs (5S, 5.8S, 18S, and 25S/28S rRNA). Biogenesis of ribosomes is meticulously regulated by integrating translation of ribosomal proteins; transcription, processing and modification of rRNAs; and assembly of these components.

Within rRNAs, the vast majority of modifications are either pseudouridylation (Ψ) or 2′-*O*-methylation (2′Ome) whose deposition is most commonly facilitated by small nucleolar ribonucleoproteins (snoRNPs). In addition to guiding the deposition of these modifications, certain snoRNPs have critical roles in the regulation the processing and maturation of rRNAs. Further integrating their role in ribosome biogenesis, many snoRNAs, the RNA component of snoRNPs, are found within the introns of ribosomal proteins or ribosome biogenesis factors. In this review, we will discuss the biogenesis of snoRNPs and their role in directing ribosome assembly.

## 2. Structure and Components

Generally, snoRNPs come in two major classes: Box C/D and Box H/ACA. Although, there are non-canonical snoRNPs, such as RMRP, that have critical roles in ribosome biogenesis, or TERC, which aids in maintenance of telomeres. Regardless of the class, each snoRNP is composed of a snoRNA and an assemblage of proteins. The nomenclature denoting each class of snoRNP is derived from conserved sequence elements within each snoRNA.

### 2.1. Box C/D snoRNAs

Box C/D snoRNPs catalyze the methylation of the 2′ hydroxyl group of ribose in RNA (Figure 1A) [5]. The RNA component of these, Box C/D snoRNAs, shares two sets of conserved sequence motifs: Box C (RUGAUGA) and Box D (CUGA) (Figure 1B). The U3 (SNORD3) snoRNA was the first to be fully characterized (discussed below). Analysis of U3 snoRNAs from humans, *Xenopus laevis*, *Xenopus borealis*, *Rattus rattus, Saccharomyces cerevisiae*, *Schizosaccharomyces pombe,* and *Dictyostelium* revealed the conservation of Box C and Box D [6,7,8]. But, it was only after the characterization of the U8 (SNORD118) and U13 (SNORD13) snoRNAs that it was appreciated that these belonged to a distinct class of small RNAs [9]. Box C and Box D exist in pairs on a single molecule, called Box C/C′ and Box D/D′, with the C′ and D′ boxes being more degenerate. Structurally, Box C/D snoRNAs are hairpins containing a large internal loop, bounded by the Box C/C′ and Box D/D′ motifs. Box C and D can base-pair with each other forming stem-bulge-stem structure called a “kink-turn” or “K-turn” motif.

In addition to Box C/D snoRNAs, K-turns are found in multiple RNA species, including mRNAs, riboswitches, and small nuclear (sn)RNAs but were first discovered and described in ribosomal RNAs [10]. A canonical K-turn is composed of two stems separated by an internal loop. The first stem, termed the “canonical stem” (C-stem) or Stem-I, ends at the internal loop with two Watson–Crick base pairs, typically G-Cs. The second helical stem, termed the “non-canonical stem” (NC-stem) or Stem-II, begins with two non-Watson–Crick base pairs, typically sheared G-A base pairs. These are maintained by long-range interactions. Loss of this base-pairing prevents localization of Box C/D snoRNAs to the nucleolus [11]. Within the loop is an unpaired U that induces a kink in the phosphodiester backbone that bends the helical axis by ~120°. The C′ and D′ boxes have a reduced ability to form a K-turn because of the sequence degeneration.

Box C/D snoRNAs associate with four evolutionarily conserved proteins: Fibrillarin (FBL)/Nop1p, SNU13(15.5K)/Snu13p, NOP58/Nop58p, NOP56/Nop56p (Figure 1C). The catalytic methyltransferase is FBL [12]. Although originally identified in the slime mold *Physarum* [13], much of the early work on FBL relied on autoantibodies from patients with scleroderma [14]. Immunoprecipitations using these antibodies identified FBL as part of an RNP that contained snoRNAs that were later characterized as Box C/D snoRNAs. Human FBL is highly similar to its yeast homolog, NOP1, and human FBL can partially rescue viability in NOP1 mutant strains [15]. Interaction with the snoRNA depends upon SNU13, formerly 15.5K, which recognized the K-turn formed by the interaction between the C and D boxes [16]. Crystallographic data of SNU13 in complex with the U4 snRNA show that it interacts almost exclusively with the purine-rich internal loop where the bulged U fits into a pocket and is stabilized by the tandem sheared G-A base-pairs [17,18]. Binding of SNU13 to this motif is essential for recruitment of other Box C/D snoRNP factors. In contrast, the sequence of stem-II of the K-turn is essential for interaction of NOP56, NOP58, and FBL, but not SNU13 [11].

The assembled snoRNP mediates site-specific 2′-*O*-methylation through antisense guide base-pairing. Early work showed that specific snoRNA species could hybridize with mature rRNA species through canonical Watson–Crick base-pairing [19], which was later extended to other snoRNA species [20,21,22,23]. This work, coupled with the observation that the hybridization occurred at sites of methylation, suggested that these interactions served as guide RNAs. Proof came shortly after that Box C/D snoRNA hybridization to rRNA directed sites of methylation [5,24,25]. This allows for a common set of proteinaceous components to be assembled on specific guide RNAs in a modular manner. This mode of regulation has become a repeated feature of evolution as it was discovered that microRNAs and CRISPR RNAs use this same strategy to determine their RNA targets. Base-pairing occurs immediately 5′ of the D and D′ boxes. This positions the rRNA to be methylated at the active site of FBL located 5 nucleotides from the 5′ end of Box D [5,26,27].

Take home message: Box C/D snoRNPs catalyze the deposition of 2′-*O* methylation using RNA-RNA base-pairing to direct target sites. FBL is the catalytic component of the Box C/D snoRNP.

### 2.2. Box H/ACA snoRNAs

Box H/ACA snoRNPs catalyze the isomerization of uridine to pseudouridine [28,29]. To generate Ψ, uridine is rotated 180° around the C_6_-N_3_ axis to generate a carbon-carbon glycosidic bond as compared to the carbon-nitrogen glycosidic bond in uridine (Figure 2A). This rotation allows for Ψ to make more hydrogen bonds by freeing up N_1_.

That Box H/ACA snoRNAs constitute a separate class of snoRNAs was recognized after the characterization of Box C/D snoRNAs. Originally characterized by the lack of the conserved C and D boxes, it was later observed that these snoRNAs contained a conserved ACA sequence at their 3′ ends [30]. Later, it was shown that these have two stem-loop structures separated by a conserved single-stranded motif that was termed the “hinge” region, or H box, with a consensus sequence of ANANNA [31] (Figure 2B). Secondary structure analysis also predicted internal loops within each stem-loop. These internal loops end up serving as the guide sequences similar to those 5′ of Box D in Box C/D snoRNAs. The uridine residue to be modified is positioned at the base of the stem-loop in a site known as the “pseudouridylation pocket”. Efficient pseudouridylation requires three structural motifs [32]: (1) Two stable hairpins containing pseudouridylation pockets, 2) approximately 15 nucleotides between the target pseudouridine and either Box H or ACA, and (3) a sufficient amount of base-pairing between the target RNA and the snoRNA. However, what constitutes a sufficient amount of base-pairing is still under investigation. The base-pairing constraints between rRNA and Box H/ACA seem to be less rigid than for Box C/D snoRNAs that adhere to a +5 base-pair rule. At a minimum, eight base-paired nucleotides are required for pseudouridylation but the flexibility of the pseudouridylation pocket allows for multiple unpaired nucleotides [33]. The higher the degree of canonical base-pairing correlates with the increased rate of pseudouridylation [34]. However, affinity does not correlate with pseudouridylation as multiple near-cognate interactions can have high affinity but reduced pseudouridylation activity.

Pseudouridylation requires the assembly of a H/ACA snoRNP composed of the snoRNA and four protein co-factors: NHP2, NOP10, Gar1, and Dyskerin (DKC1)/Cbf5 (Figure 2C). The first of these to be identified in yeast was Gar1, which was shown to be required for pre-rRNA processing and to associate with a subset of snoRNAs, which were later shown to be H/ACA snoRNA [35]. Later work showed that it was required for pseudouridylation [36]. Bioinformatic searches for proteins related to bacterial pseudouridine synthetases (PUS) identified DKC1/Cbf5 as the possible catalytic component of the Box H/ACA snoRNP, which was confirmed experimentally [37,38]. Chromatographic separations and affinity purifications identified Nhp2 and Nop10 as the final components of the Box H/ACA snoRNP [38,39]. Nhp2 is not required for guide RNA binding, but is absolutely critical for efficient pseudouridylation [40].

Take home message: Box H/ACA snoRNPs catalyze the conversion of uridine to pseudouridine using RNA-RNA base-pairing to direct target sites. DCK1/Cbf1 is the catalytic component of the H/ACA snoRNP.

## 3. Transcription and Processing of snoRNAs

SnoRNAs are found in different genomic contexts that broadly fall into two sub-dividable categories: Independent transcription units and intronically encoded snoRNAs (Figure 3). In yeast and plants, most snoRNAs are found as independently transcribed genes, either as monocistronic or polycistronic units. The former is more common in yeast, while the latter is more common in plants. Some metazoan snoRNA genes exist as independent genes, but these are among the most essential (e.g., U3, U8, and RMRP). The majority of the information regarding the promoters of independent snoRNAs genes comes from yeast, but this data is still lacking. Many yeast snoRNA genes drive transcription from TATA-containing promoters downstream of an A/T rich region and a Rap1 binding site [41]. Rap1 is a transcription factor that is also necessary for transcription of ribosomal protein genes in yeast, which may serve to integrate these processes in ribosome biogenesis [42]. Subsequently, Tbf1 was identified as a second transcription factor necessary for snoRNA transcription [43]. In humans, the structure of independently transcribed snoRNA promoters is largely unknown although some progress has been made through bioinformatic analysis [44].

While RNA polymerase II (Pol II) transcribes the majority of independently transcribed snoRNA genes, there are some important exceptions. In yeast, all snoRNAs are transcribed by RNA Pol II except snR52, which is transcribed by RNA Pol III [45,46,47]. In metazoans, more independently transcribed snoRNA genes are transcribed by RNA polymerase III. In *Caenorhabditis elegans*, as many as 59 snoRNA genes are transcribed by RNA polymerase III [48] and at least two in *Drosophila* [49]. Of particular note are some homologous snoRNA genes that have switched polymerases throughout evolution. The U3 snoRNA, a critical snoRNA necessary for rRNA maturation (see below), is transcribed by RNA polymerase II in yeast and in humans [50,51,52]. Yet, in plants, the U3 snoRNA is transcribed by RNA polymerase III [53]. In humans, RMRP, a non-canonical snoRNA is transcribed by RNA polymerase III [54], but in yeast its homolog, NME1, is transcribed by RNA polymerase II [55].

Following transcription, pre-snoRNAs must be processed to generate a mature snoRNA (Figure 3A). This was recently covered in an excellent review [56], so we will only cover the basics here. Since most yeast snoRNAs are transcribed as independent transcription units, the majority of the data regarding the maturation of this class of snoRNAs comes from yeast. Transcription termination and 3′ end formation is facilitated by the Nrd1-Nab3-Sen1 (NNS) complex and Pcf11 [57,58]. Following cleavage, the 3′ end is exonucleolytically trimmed by the exosome to generate the mature 3′ ends [59]. Initially, the 5′ end is capped with a canonical m^7^G cap. However, this is either removed by Rnt1 [60], a member of the RNase III family, or converted to a m^2,2,7^G trimethylguanosine cap by TGS1 [59]. If the cap is removed, the 5′ end is trimmed by the exonuclease Rat1 and Xrn1 [61]. Cleavage by Rnt1 occurs co-transcriptionally and facilitates efficient 3′ end processing [62].

Higher eukaryotes have largely embraced a more nuanced mechanism of snoRNA transcription (Figure 3B). Rather than being transcribed as individual genes, metazoan snoRNA genes are largely found within the introns of host genes. The presence of the three copies of the U14 (SNORD14) snoRNA localized to the introns of the hsc70 gene was discovered in humans, mice, and rats [63]. Later, it was shown that this was processed from these snoRNAs and could be processed from the introns of pre-mRNA in a *Xenopus* injection system [64]. Later work showed that this was not a unique situation as the U15 (SNORD15) snoRNA was identified to be localized within an intron of the RPS3 gene and processed from its pre-mRNA [65]. The discovery of a ribosomal protein gene hosting a snoRNA portended the discovery that many intronically encoded snoRNAs are found within genes encoding ribosomal biogenesis or translational machinery. For example, more than a third of ribosomal protein genes host at least one snoRNA. Some ribosomal protein genes host multiple snoRNAs, such as RPL7A or RPL13A, which hosts four snoRNAs each. Theoretically, this arrangement allows for coupled synthesis of ribosomal proteins and snoRNAs which are required for modification of rRNA. An interesting case of coregulation has been shown for NOP56 and snoRD86 [66]. NOP56 is a component of the Box C/D snoRNP complex that has been proposed to function in “locking in” the final conformation of snoRNPs [67]. When NOP56 protein levels are high, free NOP56 binds to the SNORD86 in the intron of the NOP56 pre-mRNA and alter splicing to generate a NOP56 mRNA with a premature termination codon. Thus, this isoform of NOP56 is degraded via nonsense-mediated decay (NMD). Alternatively, when NOP56 levels are limiting, SNORD86 is unbound and NOP56 pre-mRNA is spliced to generate an mRNA that encodes a full-length protein.

Another important set of host genes are those that appear to have no coding potential and also do not appear to function as long noncoding (lnc)RNAs. Among the most noteworthy is the U22 host gene (UHG), which, in addition to U22 (SNORD22), hosts seven additional snoRNAs (U25–U31) [68]. All snoRNAs in UHG are Box C/D snoRNAs. After splicing, the UHG mRNA transiently associates with ribosomes but is then rapidly degraded by nonsense-mediated decay (NMD), suggesting that it has no purpose other than to serve as a host to these eight snoRNAs. A second member of this class was discovered soon after called *gas5*, which hosts 10 snoRNAs (nine in mouse) of the Box C/D family [69]. Similar to UHG, GAS5 mRNA is rapidly degraded in normal growing cells. However, GAS5 mRNA is stabilized in multiple different cancer types and appears to have functions distinct from its snoRNA encoding activity [70]. Subsequently, other host genes, such as U17HG [71], U19HG [72], and U50HG [73], have been identified. Strikingly, all of these host mRNAs contain 5′ terminal oligopyrimidine (5′TOP) motifs. These motifs are thought to regulate translation through an mTOR-dependent mechanism [74]. What is worthwhile to note here is that all ribosomal proteins are encoded by mRNAs that contain 5′TOP motifs, potentially providing a link between different classes of snoRNA host genes.

Again the molecular intricacies of snoRNA excision from introns has been covered in depth recently [56], so we will only cover basics here. The basics of processing are similar to that of independently transcribed snoRNA genes: After 5′ and 3′ processing, final maturation takes place by exonucleolytic trimming. However, this can occur via two pathways: A splicing-dependent and splicing-independent pathway [75]. The splicing-dependent pathway appears to be the major pathway. Here, the splicing machinery removes the intron leaving a lariat containing the pre-snoRNA. In this pathway, maturation depends upon linearization by the debranching enzyme [76]. At this point, 5′ to 3′ and 3′ to 5′ exonucleases trim to generate the mature 5′ and 3‘ ends [77]. The second, minor pathway does not rely on intron excision followed by debranching. Instead, a stem structure is formed at the base of the pre-snoRNA that is cleaved, releasing the snoRNA from the larger precursor [75]. Unlike independently transcribed snoRNA genes, intronically encoded genes lack a 5′ cap, instead of containing a 5′ monophosphate generated after exonucleolytic trimming.

Assembly of both Box C/D and Box H/ACA snoRNPs require multiple assembly factors that mediate RNP formation and trafficking from the site of transcription, through Cajal bodies and to the nucleolus. This complex process has recently been reviewed in [78]. However, one complex of note is the HSP90/R2TP chaperone system. Originally discovered in yeast via proteomic analysis, the R2TP complex is composed of Rvb1, Rvb2, Tah1, and Pih1 and interacts with HSP90 [79]. The identification of the human complex, containing homologs RUVBL1, RUVBL2, RPAP3 and PIH1D1, demonstrating that this is a conserved complex [80]. Despite having unrelated RNA structures and proteinaceous components, this complex aids in the assembly of both Box C/D and Box H/ACA snoRNPs [81]. Depletion of RUVBL2 leads to a loss of Box C/D and Box H/ACA snoRNAs and to a defect in the trafficking of snoRNP proteins to the nucleolus [82].

Take home message: Genomically, snoRNAs are found as independent transcription units or hosted within introns of mRNAs. Both require post-transcriptional processing and assembly with proteinaceous components.

## 4. Role of Modifications in Ribosome Function

In many cases, the roles of rRNA modifications remain enigmatic. The conservation and clustering of modification at functional centers, such as the peptidyl transferase center or the subunit interface, point to their importance. With regards to ribosome structure and stability, 2′-*O*-methylation increases hydrophobicity and pseudouridylation allows for additional hydrogen bonding. Both of these properties can stabilize interactions or provide rigidity. In yeast, global loss of either 2′Ome or Ψ generated through the expression of catalytically dead mutants of Nop1 (FBL) and Cbf5p (DKC1) results in severe phenotypes [12,83]. Inactivating mutations to DKC1 in mammals (D95A) expectedly prevents pseudouridylation and reduces translation rates and fidelity [84]. However, preventing individual modifications through abrogation of a single snoRNA rarely has dramatic effects [85]. However, there are some of note. In yeast, snR10 catalyzes the conversion of U_2923_ to Ψ_2923_ in the 25S rRNA. Deletion of this snoRNA renders cells more susceptible to osmotic and cold stresses [86]. However, slightly complicating this data is that loss of this snoRNA also reduces the efficiency of pre-rRNA processing (discussed below) [87]. Yet, there is an observed decrease in translation efficiency that is more likely tied to modification rather than processing [88]. Again, in yeast, loss of methylation of U_2918_ in the 25S rRNA reduced growth and exhibited translational defects [89]. The importance of this methylation is revealed by the fact that it is deposited by two redundant mechanisms, snR52- and spb1-mediated methylation. Additionally, methylation of the equivalent site in *Escherichia. coli* is required for optimal fitness [90]. However, the importance of these individual modifications appears to be the exception, rather than the rule. Yet, the fact that snoRNAs, components of the snoRNPs, and sites of modification are under selective pressure validates their importance. Further work demonstrated that rather than acting individually, many of these modifications have a cumulative and synergistic effect on ribosome function. Loss of six Ψ in the peptidyl transferase center of the large subunit greatly reduces translation rates and polysome assembly [88]. An interface between the large and small subunits occurs between Helix 69 of the large subunit and helix 44 of the small subunit [91]. Helix 69 contains multiple modified nucleotides over an 11-nucleotide stretch (AmCΨAΨGACCΨCΨ). Abrogation of these modifications through disruption of snoRNAs impairs cell growth, disrupts ribosome structure, and alters translational efficiency [92]. A cluster of snoRNP-deposited modifications in the decoding center of the large subunit is also required for efficient translation [93].

Take home message: Multiple chemical modifications of rRNA have a cumulative effect on ribosome activity.

## 5. Pre-rRNA Processing

While most snoRNPs function in rRNA modifications, a subset directs rRNA processing. While loss of a single snoRNA that guides modification is rarely detrimental, the loss of a snoRNA that directs processing is often lethal. The rRNA maturation is a baroque process, requiring multiple endonucleolytic cleavage and exonucleolytic trimming reactions (Figure 4). Ribosomes contain four mature rRNAs, yet they are generated from two precursor molecules. The pre-5S rRNA is transcribed by RNA polymerase III and requires exonucleolytic trimming to generate the 5′ and 3′ ends. The remaining rRNAs are generated from long polycistronic precursors called the 35S rRNA in yeast or 47S rRNA in mammals. These contain the 18S, 5.8S, and 25/28S rRNA separated by two internal transcribed spacers (ITS1 and ITS2) and flanked by two external transcribed spacers (5′ETS and 3′ETS). Within these spacers are multiple sites that are targeted by nucleases that mature the initial precursor into mature rRNAs. The majority of these processing events occurs in the nucleolus, although final maturation occurs in the cytoplasm.

## 6. The snoRNPs’ Role in rRNA Processing

### 6.1. U3 (SNORD3) snoRNP

The U3 snoRNP is a member of the Box C/D family that typically facilitates 2′-*O*-methylation of ribose sugars (Figure 5A). However, U3 has not been shown to guide any chemical modifications. Instead, the U3 snoRNP is essential for the formation of the small subunit (SSU) processome, a multi-subunit complex that directs maturation of the 18S rRNA and 40S small subunit.

The U3 snoRNA was first identified by fractionation of RNA from the rat liver [96]. Further fractionation of the uridine-rich 7S fraction identified RNA species that were termed U1, U2, and U3 [97]. This RNA was soon shown to associate with ribosomes, giving the first hints at its possible function in cells [98,99]. Like other independently transcribed snoRNAs, the U3 contains a trimethylguanosine (m^2,2,7^G) cap [100]. Cloning of the human and rat U3 snoRNA genes significantly increased progress on understanding the role of this abundant RNA [101,102].

The fact that the U3 snoRNA will interact with rRNA in deproteinized cell extracts suggested that it interacted with these RNAs through base-pairing [98]. Using psoralen-assisted UV crosslinking, the U3 snoRNA was shown to crosslink to RNA contained within the nucleolus in cell [103]. These techniques were refined and it was later shown that the U3 snoRNA crosslinked in the 5′ external transcribed spacer, immediately downstream of the primary rRNA processing site, termed A′/01, in mammalian cells [104]. Later, U3 snoRNA base-pairing to the 5′ETS was shown to be conserved in yeast [105]. The 5′ end of the U3 snoRNA can fold into a hairpin in isolation (Figure 5A), but the 5′ end of the snoRNA serves as the main platform for base-pairing with the pre-rRNA and thus this hairpin must unfold (Figure 5B). The length of this 5′ extended regions interacts with various points of the 5′ETS and 18S rRNA in which the intervening sequences are looped out, helping guide the formation of the ribosome [106].

Several studies proposed that U3 was involved in rRNA processing [107,108,109,110]; however, conclusive proof that U3 is necessary for processing came from the labs of Joan Steitz and Barbara Sollner-Webb in 1990 [111]. Micrococcal nuclease treatment of nuclear extracts inhibited the processing of an rRNA substrate in vitro, while DNase I treatment had no effect, thereby establishing the necessity for an RNA co-factor. Through several rounds of immunodepletion using antisera from scleroderma patients against FBL, a component of the U3 snoRNP, they were able to deplete nuclear processing extracts of U3 snoRNA. Upon depletion, these extracts failed to process rRNA. Finally, to demonstrate that U3 was involved and not another FBL-bound snoRNA, the authors demonstrated that oligonucleotide-directed, RNase H-mediated cleavage of U3 similarly blocked rRNA processing. Shortly afterwards, the necessity for U3 in rRNA processing was shown in living cells after the injection of U3 blocking oligos into frog oocytes [112]. Later, the necessity of U3 snoRNA in rRNA processing was shown in yeast by depleting the U3 snoRNA through use of a galactose inducible promoter [113].

While the snoRNA itself and base-pairing of the snoRNA to the 5′ETS of the rRNA precursor is required for activity, it is not sufficient for cleavage. Assembly of the snoRNA into a snoRNP is necessary for activity. Autoantibodies against FBL coprecipitated the U3 snoRNA. In yeast, depletion of FBL was shown to have the same effect on rRNA processing as depletion of the U3 snoRNA, suggesting that the interaction between the two is required for activity [114]. The U3 snoRNP contains all canonical members of Box C/D snoRNPs (i.e., SNU13 (15.5K), NOP56, and NOP58) [110]. However, the uniqueness of the U3 snoRNP is highlighted by its additional protein components. One of the most prominent additional factors is called hU3-55K in humans or Rrp9 in yeast. This protein was first identified in Chinese hamster ovary cells through affinity chromatography [115]. Similar to other components of the U3 snoRNP, U3-55K is required for viability and rRNA processing [116]. More recently, Clerget and colleagues showed that yeast pre-rRNA processing defect was significantly enhanced when specific regions of the Rrp9 and U3 snoRNA were mutated [117]. The locations that each protein binds to the U3 snoRNA were identified using a technique known as crosslinking and analysis of cDNA (CRAC) [118]. Two other components were identified, known as Sof1 and Mpp10 [15,119].

The identification of Mpp10 proved particularly fruitful in the next major discovery regarding the U3 snoRNP. The U3 snoRNP is a central component of a large processing complex called the small subunit (SSU) processome [120]. Using tandem affinity purification of Nop5/58, the yeast homolog of NOP58, to first purify all Box C/D snoRNAs followed by Mpp10 to purify U3 snoRNP complexes, Dragon et al. showed that the U3 snoRNP associated with at least 28 in a complex of approximately 2.2 MDa. They termed the previously unknown proteins “U-three proteins” or UTPs. Depletion of any of these proteins blocked the formation of the 18S rRNA. Finally, this led to the identification of the “terminal knobs” first seen by Oscar Miller in chromatin spreads of rDNA repeats in 1969 [121]. The Baserga lab determined that these terminal knobs were the SSU processome through depletion of these newly identified UTP proteins. It is worth noting that these terminal knobs had previously been suggested to contain rRNA processing complexes [122]. This discovery along with major advances in cryoelectron microscopy led to the first structures of the SSU processome, alternatively called the 90S pre-ribosome [106,123,124]. More recently, the high-resolution structure of the SSU has been solved in yeast [125] and in the thermophilic fungus *Chaetomium thermophilum* [126].

Take home messages: The U3 snoRNP is required for small subunit assembly.

### 6.2. RNA Component of RNase MRP Complex (RMRP)

RMRP is a non-canonical snoRNA that belongs to neither the Box C/D nor Box H/ACA classes. That RMRP has a role in human pre-rRNA processing has long been established, but the precise role was not understood until recently. Only with the generation of CRISPR/Cas9-mediated deletions of the RMRP gene was it definitively shown that it directs the cleavage at site 2 in ITS1 of human pre-rRNA [127]. RMRP is the RNA component of the RNase MRP (ribonuclease mitochondrial RNA processing) complex. The RNase MRP endonuclease was originally identified in mouse, showing its ability to cleave the mitochondrial RNA that functions as a primer for mitochondrial DNA replication [128]. RMRP contains a decamer sequence, 5′-CGACCCCUCC-3′, complementary to a conserved sequence adjacent to the enzymatic cleavage site on the mitochondrial RNA substrate, and is present in the RNase MRP RNA [129,130,131]. Despite initially being presumed to be active in the mitochondria, RMRP is nuclear encoded. Human RMRP is 267 nucleotides long and shares 84% homology to the corresponding mouse gene; surprisingly, at least 700 nucleotides of the immediate 5′-flanking region are conserved [132]. The sequence of the RMRP transcript is highly conserved among a variety of different species, including human, mouse, rat, cow, Xenopus, yeast, Arabidopsis, and tobacco [133]. The length of the transcript varies among different species. Secondary structure models for RMRP reveal a complex structure, the core of which is required for the assembly and function of the ribonucleoprotein complex [134,135]. It has been shown that the MRP RNA of human cells is identical to the “Th” or 7-2 RNA originally described as being precipitable with autoimmune sera of scleroderma patients [136]. Since this species (MRP/Th) is known to be present in the nucleolus [137,138], it is most likely that RMRP could play an essential role in the metabolism of rRNA. More evidence that came from the immunolocalization data suggested that RMRP may be involved in stages of ribosome biogenesis in the nucleolus [139]. The apparent predominant localization of RMRP to the nucleolus suggests that RNase MRP might somehow be involved in nuclear rRNA processing.

In *Saccharomyces cerevisiae*, the RMRP ortholog NME1 (nuclear mitochondrial endonuclease 1) showed an essential role in cell viability, indicating a critical nuclear role for RNase MRP [55]. MRP cleaves the pre-rRNA at the A_3_ cleavage site in yeast pre-rRNA, which is thought to be the functional equivalent of site 2 in humans [140]. Further, conditional depletion of RNA component of the enzyme showed that this is responsible for the maturation of 5.8S rRNA. It was found that there was a reversal in the stoichiometry of the two mature forms of 5.8S rRNA; in the MRP RNA-depleted condition, the 7-nucleotide-longer version of 5.8S rRNA was 10 times more abundant than the shorter species lacking this 7-nucleotide sequence at the 5′ end. These results were in contrast to the normal stoichiometry in which the shorter version of 5.8S rRNA is 10-fold more abundant than the slightly longer version. It is worth mentioning that a particular A to G transition at position 122 at RNA sequence defines its functional capacity [141,142,143]. High-copy suppressor analysis of this point mutation leads to identification of interacting protein and it was shown that SNM1 protein is the first identified protein component unique to the RNase MRP enzyme complex. The protein contains a leucine zipper motif, a zinc-cluster motif, and a serine/lysine-rich tail [144]. Salinas et al. identified another protein component of this endonuclease and named it *RMP1*, for RNase MRP protein [145]. Characterization of the entire RNase MRP endoribonuclease is found to be complicated by the fact that eight of the known proteins of the complex are shared with a related ribonucleoprotein, called RNase P. RNase P is also an endoribonuclease but it is mainly involved in tRNA precursors’ maturation [146].

Another role has been assigned to this RMRP by observing a delay in the progression of the cell cycle at the end of mitosis in some *nme1* mutants [147]. This is caused by an increase in *CLB2* (B-type cyclin) mRNA levels leading to increased Clb2p (B-cyclin) levels and a resulting late anaphase delay. One reason for the cell cycle delay in these mutants might be the increased level of CLB2 mRNA. Normally, the RNase MRP complex cleaves the 5′ UTR of CLB2 mRNA. That, in turn, causes a rapid degradation of CLB2 mRNA and efficient cell cycle progression [148].

Take home message: RMRP is a non-canonical snoRNA that is involved in processing of ITS1.

### 6.3. Other snoRNAs Necessary for rRNA Processing

The U3 and RMRP are the two-best characterized snoRNPs that are necessary for rRNA maturation, but they are not unique. The U8 and U13 snoRNAs are also independently transcribed snoRNAs genes and are necessary for rRNA processing [68]. Both of these are members of the Box C/D family of snoRNPs. U8 appears to be a vertebrate-specific snoRNA that is required for the maturation of the 5.8S and 28S rRNAs [149]. Disruption of U8 snoRNP function in *Xenopus* results in failure to mature large subunit rRNAs. U8 is also necessary for large subunit processing in mice and was recently shown to be necessary in humans as well [150,151]. The critical nature of this snoRNA in vertebrates raises the question as to how yeast survive without this snoRNA. It has been proposed that portions of yeast ITS2 can adopt two different secondary structures, termed the “ring model” and “hairpin” model, that can independently guide processing events in the absence of the U8 snoRNP [152]. Upregulation of U8 is commonly seen in breast cancers and has been proposed to be a potential cancer biomarker [151].

The U13 snoRNA was among the earliest discovered snoRNAs [9]. Early work demonstrated that U13 was required for pre-rRNA processing through Watson–Crick base-pairing to the 3′ end of the 18S rRNA [153]. Despite this, the precise role of this snoRNA in pre-rRNA processing has remained enigmatic, but recent data raise some intriguing possibilities. The Suzuki lab found that the biogenesis of the 18S rRNA was dependent upon *N^4^*-acetylation of C_1773_ (ac^4^C_1773_) in yeast. Ultimately, they showed that this was dependent upon yeast acetyltransferase KRE33/RRA1. The human homolog of this protein, NAT10, performs the same function at C_1842_ of the human 18S rRNA [154]. Further work suggested that this is also responsible for acetylation of C_1337_ of the 18S rRNA in humans and C_1297_ of the 18S rRNA in yeast [155]. But the more surprising result from Sharma et al. (2015) was that the U13 snoRNP was essential for deposition of this acetylation. In yeast it was shown that snR45 was the functional homolog of vertebrate U13 which guides acetylation of C_1773,_ whereas snR4 directs acetylation of C_1280_. As U13 has not been shown to direct any 2′-*O*-methylations, it is intriguing to believe that this snoRNA has evolved to direct acetylation that is required for processing.

U14 has undergone intense study over the last 30 years and is also necessary for pre-rRNA processing although this may be linked to its ability to direct 2′-*O*-methylation. Like U3, U8, and U13, the U14 snoRNA is a member of the box C/D family, but unlike those, this snoRNA is intronically encoded, rather than being independently transcribed [63]. In yeast, depletion of the U14 results in misprocessing of the 18S rRNA [156]. Like other snoRNAs, U14 base-pairs with pre-rRNA which, in this case, is essential for maturation of the 18S rRNA [19,157]. This is also true in yeast [156]. However, yeast U14 is distinguished from its vertebrate homologs by the presence of a stem-loop domain that is essential for its function in pre-rRNA processing. This element, known as the Y-domain, for “yeast specific”, is located between the regions that base-pair with rRNA [158,159]. Unlike U3, U8, or U13, the U14 snoRNP also functions in its canonical role as a Box C/D snoRNP by directing methylation of the 18S rRNA [160]. It is unclear if U14-dependent 2′-*O*-methylation is required for processing or if this is a separate function of the snoRNP.

In addition, several other snoRNPs have been shown to be necessary for efficient pre-rRNA processing; however, the precise mechanism by which they perform these functions is unknown. These include the U22 (SNORD22) [161], U17 (SNORA73)/snR30 [162,163], SNORA62 (E3), SNORA63 (E3) [164], and snR10 [87].

Take home message: Various snoRNAs have been implicated in processing of rRNA, but the precise mechanism by which they perform this function has yet to be described.

## 7. Role of snoRNP-Mediated Modifications in rRNA Processing

It is clear that snoRNAs are necessary for rRNA processing, particularly those discussed above (e.g., U3, RMRP). Indeed, even those that guide RNA modification have been shown to have a role in rRNA processing (e.g., snR10, U13, U14). However, showing that a particular snoRNA is required for processing does not distinguish between two possibilities: (1) That snoRNP:rRNA interaction is required for processing or (2) that chemical modification of particular nucleotides is necessary for processing. That is, it is possible that certain snoRNPs act as chaperones, changing the secondary structure of the pre-rRNA, which promotes efficient cleavage. Therefore, the snoRNP:rRNA interaction could serve to chemically modify the rRNA and alter the rRNA secondary structure.

Early work by the Darnell lab presented data demonstrating that 2′-*O*-methylation of ribosomal RNA was essential to pre-rRNA processing, but not for rRNA transcription [165]. Later work, using ethionine, a methionine analog that is not capable of acting as a methyl donor, confirmed these results [166]. Using cycloleucine, another inhibitor of methylation, others suggested that loss of 2′-*O*-methylation delayed or reduced the efficiency of processing, but did not completely abolish it [167]. Surprisingly, while 2′-*O*-methylation is evolutionarily conserved, the need for 2′-*O*-methylation in rRNA processing may not be. In *E. coli*, treatment with ethionine appears to have no effect on rRNA processing [168]. However, while rRNA maturation is unaffected, 50S subunits from *E. coli* grown in ethionine were defective, but this appears to be due to lack of methylation of ribosomal proteins. The fact that rRNA methylation is not required for assembly of *E. coli* ribosomes may not be surprising since *E. coli* ribosomes can be assembled in vitro with submethylated rRNA [168]. Further, it is worth pointing out that 2′-*O*-methylation in *E. coli* is not meditated by snoRNAs. More surprising is that in yeast, some temperature-sensitive mutants in Nop1p/FBL that affect methylation do not impair rRNA biogenesis, but do prevent 2′-*O*-methylation [12]. Therefore, it seems as if higher eukaryotes are unique in the necessity of 2′-*O*-methylation for maturation of rRNA.

It has been more difficult to determine the role of Ψ in rRNA processing due to the lack of inhibitors of pseudouridylation, particularly in vertebrates; however, some progress has been made. In yeast, Cbf5p is essential for viability [169]. However, expression of catalytically dead (D95A) Cbf5p in ΔCbf5p cells can rescue viability, but these cells lack Ψ in rRNA [83]. Despite this, there was no apparent lack of effect on rRNA processing. Similar to yeast, DKC1 knockout causes embryonic lethality in mice [170], but MEFs, cells carrying catalytically dead (D125A) mutations, show defects in rRNA processing and synthesis [171]. These data are somewhat complicated by reduced levels of DKC1 in these cells. Regardless, the authors did find that mature rRNA from DKC1(D125A) cells were less stable, arguing that Ψ has a role in stabilizing rRNA. In total, these data mirror that of 2′-*O*-methylation in which Ψ is required for processing in higher eukaryotes, but appears to be more dispensable in yeast. However, as mentioned above, while unmodified rRNA can be matured, the ribosomes they comprise are defective.

There are instances, such as the above-mentioned snR10, in which snoRNAs that direct modifications are necessary for processing [86,87]. Yet it has not been firmly established that modification is required or snoRNA:rRNA hybridization is required for processing. Also, pseudouridylation by snR35 converts U_1191_ to Ψ_1191_ which is further modified to 1-methyl-3-γ-(α-amino-α-carboxypropyl)-pseudouridine (m^1^acp^3^Ψ) [172]. This modification is required for normal processing of yeast 18S rRNA. Failure to convert U_1191_ to Ψ_1191_ prevents deposition of m^1^acp^3^. As a result, the 20S pre-rRNA never matures to the 18S rRNA [93].

Take home message: There are specific instances in which chemical modification of rRNA aids in pre-rRNA processing, but most are dispensable. The trend is that these modifications are more required in higher eukaryotes than in yeast or bacteria.

## 8. Diseases

Loss of specific snoRNAs or reduction in the activity of catalytic components of snoRNPs have been tied to multiple disease states including congenital disorders and cancer. Here, we only focused on those involved in the regulation of ribosome biogenesis or activity, but it is worth mentioning that there are certain instances, such as deletion of SNORD116, which causes Prader–Willi syndrome, but appears unconnected to ribosome function. Recent work also shows that certain snoRNAs can direct modification of mRNAs and tRNAs that likely have a role in maintaining cellular homeostasis [173,174].

Cartilage-hair hypoplasia (CHH) is a form of dwarfism, first identified by Victor McKusik in Amish communities [175]. Patients with CHH are characterized by a short stature, hypoplastic hair, and short limbs. Mutations in the RMRP snoRNA were shown to be causative for CHH [176]. Subsequent work has revealed nearly 100 different mutations in the RMRP gene related to CHH (reviewed in [177]). Disease-associated mutations in RMRP disrupt normal rRNA processing [178], as does complete ablation of RMRP gene by CRISPR-Cas9 [127].

Another prominent disease related to snoRNP function is X-linked dyskeratosis congenita (X-DC). Classical X-DC is characterized by hematopoietic defects, such as bone marrow loss, and cutaneous abnormalities, such as abnormal pigmentation [179]. Mutations in DKC1 are prominent in X-DC, but non-X-linked mutations have also been found in other Box H/ACA components, NOP10 and NHP2 [180,181,182]. As components of the Box H/ACA snoRNPs, these factors are all necessary for pseudouridylation of rRNA; however, the precise molecular defect underlying X-DC has been controversial [183,184]. This is because these factors also associate with a non-canonical Box C/D snoRNA called telomerase RNA component (TERC), which is required for telomere maintenance [185]. Indeed X-DC patients have shorter telomeres which surely have a role in disease progression [186]. Yet, patients also see a defect in rRNA modification and translation. Mouse models of X-DC show a marked decrease in rRNA modifications [187]. These same models show defects in translation, particularly on those mRNAs that harbor internal ribosome entry sites (IRES) that facilitate cap-independent translation initiation [188]. These include alterations to the translation of the p53 mRNA, which may explain the increased susceptibility to cancer in X-DC patients [189]. X-DC-associated mutations in DKC1 reduce the basal level of particular snoRNAs, thereby reducing rRNA pseudouridylation [190]. These defects impair hematopoietic stem cell differentiation, likely leading to bone marrow failure. The linkage between hematopoietic defects in X-DC is part of a recurring theme in ribosome biology. Other defects in ribosome biogenesis, not necessarily related to snoRNPs, are termed “ribosomopothies” and often present in patients as anemias or other defects in hematopoiesis [reviewed in [191]]. These include Diamond-Blackfan anemia, Shwachman–Diamond syndrome, and Treacher–Collins syndrome. Recently, it was also shown that nucleophosmin (NPM), a prominent phosphoprotein in the nucleolus, is mutated in certain forms of dyskeratosis congenita [192]. Rather than causing defects in Box H/ACA snoRNPs and pseudouridylation, these mutations affect Box C/D snoRNPs and alter 2′-*O*-methylation of rRNA.

In recent years, it has also become clear that dysregulation of snoRNPs and rRNA modification is tied to oncogenesis. Much work on miRNA expression used snoRNAs as reference genes for qRT-PCR studies, until it was discovered that snoRNAs are misregulated in tumors [193]. The p53 is a direct regulator of FBL transcription and, therefore, a reduction or impairment of p53 results in increases in FBL levels [194]. Increased expression of FBL results in a concomitant increase of rRNA synthesis, but only slight changes to global rRNA methylation. Instead, there were dramatic changes to the pattern of methylation with some sites being increased five-fold and others being reduced by half. These changes to methylation altered the translational fidelity of ribosomes and increased the translation of mRNAs harboring IRES elements. These data parallel other data showing that decrease in pseudouridylation connected to X-DC reduced IRES-dependent translation [188]. The increase in FBL levels has been confirmed in mouse models of breast cancer that also show a marked increase in snoRNA levels [195]. In contrast to this data, it has been reported that acute leukemia is often tied to a global decrease in snoRNA levels [196]

While global increases in snoRNAs and FBL activity are tied to cancer development, there are also cases of specific snoRNAs being decreased in cancer. The snoRNA host gene GAS5 is significantly downregulated in breast cancer [197]. The U50 snoRNA is also hosted in a noncoding RNA that is mutated in B-cell lymphomas [73]. Subsequent work showed that U50 (SNORD50) was also misregulated in prostate and breast cancer [198,199]. Ectopic reintroduction of U50 is able to repress some of the cancer-associated phenotypes in breast cancer cells. U50 catalyzes the methylation of C_2848_ and possibly G_2863_ in the 28S rRNA. However, it is also possible that U50 has a role in directly regulating the expression of KRAS [200]. One of the more recent surprising findings is that loss of a single snoRNA, SNORA24, synergizes with oncogenic RAS expression to promote cancer [201]. In this instance, SNORA24 guides conversion of U_609_ and U_863_ to Ψ_609_ and Ψ_863_, respectively. Particularly important here is Ψ_609,_ which is found in the decoding center of the ribosome. Loss of these modifications had little effect on global translation but were defective in tRNA selection and ribosome translocation. Further investigations have identified dozens of additional snoRNAs whose expression is significantly altered in tumors, either positively or negatively (reviewed in [202]). The mechanistic consequences of many of these misregulated snoRNAs are yet to be determined.

## 9. Future Considerations

Despite the fact that snoRNPs have been under intense study for 40 years, there is still much to be learned. Recent cryo-EM data have suggested the existence of several unknown rRNA modification [2]. There remain several “orphan snoRNA” that have not yet been shown to direct Ψ or 2′Ome of rRNA or have a role in pre-rRNA processing. Given the recent discovery that some snoRNPs can direct acetylation of rRNA, a previously unknown role, it raises the possibility that snoRNPs may have a role in these yet-to-be-characterized modifications. However, it is worth pointing out that recent mass spectrometric analysis of rRNA were not able to confirm or identify these possible modifications [3]. These modifications would first need to be identified before proceeding with the investigation of what role snoRNPs have in this process.

Within the last decade, it was discovered that mature tRNAs serve as precursor molecules for smaller RNA species that have been designated by various names, such as tiRNAs, tRFs, and tRNA halves [203,204,205,206]. These fragments have various roles in biology, including regulating the epigenetic state of chromatin, global translation regulation, and miRNA-like mRNA silencing. At nearly the same time as tRNA-derived RNAs were discovered, it was also observed that snoRNAs are processed into smaller stable RNA species [207]. These RNA species were given the name snoRNA-derived RNAs (sdRNAs). Dozens of studies have confirmed the presence of these smaller RNA species. Yet, despite the progress made on characterizing the mechanisms through with tRNA-derived RNAs function, elucidating the molecular function of sdRNAs has lagged behind. Some data suggests that some sdRNAs may regulate mRNAs’ splicing [208] and recent data point to the fact that they may have a role in regulating translation, similar to some tRNA-derived RNAs [209]. Whether this occurs through similar mechanisms to tRNA-derived fragments is unknown. However, YB-1, a protein that interacts with tRNA fragments to control their activity [210], also interacts with snoRNAs [211]. This protein has multiple cellular roles but is most actively involved in mRNA translation (reviewed in [212]). However, given the immaturity of the data surrounding these RNAs, it is equally possible that the sdRNAs regulate the activity of YB-1 as the converse.

An even less appreciated class of small RNAs are those derived from rRNAs, which were also discovered at the same time as tRNA-derived RNAs and sdRNAs [203]. Recent data confirms that they are abundant and that their biogenesis may be regulated [213]. However, little is known about their activity. Given that some snoRNPs direct cleavage of pre-rRNAs, it is possible that they also play a role in the biogenesis of these small RNAs. However, a more likely connection is through their ability to modify rRNA. The tRNAs are the only RNAs that are more heavily modified than rRNAs and differential modification of tRNAs has a clear role in directing the biogenesis of tRNA-derived RNAs (reviewed in [213]). By analogy, it is conceivable that modification of rRNAs by snoRNPs could similarly direct the biogenesis of rRNA-derived RNAs.

That differential modification of rRNAs may play a role in the generation of rRNA-derived RNAs leads to the most provocative current topic with regards to snoRNPs: Their potential role in the generation of “specialized ribosomes”. While the possibility of such ribosomes has often been speculated, the discovery that mRNAs from *Hox* genes depends upon ribosomes with a precise set of associated proteins has accelerated research in this area [214]. As a basis for specialization, there must be clear evidence of heterogeneity in the ribosome pool. Work has progressed in this area by analyzing the protein complement of ribosomes. However, recently, it has become clear that rRNA modifications provide another, perhaps more prominent, layer of heterogeneity in ribosomes, that may lead to the identification of “specialized ribosomes”. That snoRNPs are responsible for the deposition of these modifications makes them important players in this process. The development of new techniques have provided many insights into this possibility [215]. Initially, ribose methylation patterns in rRNAs were identified by labeling RNA with ^32^P-orthophosphate and ^14^C-methyl-methionine and conducting RNA fingerprinting [216,217]. Pseudouridylation patterns were mapped by RNA hydrolysis followed by thin layer chromatography [218]. More site-specific techniques were developed using CMCT modification followed by analysis of reverse transcription (RT) stops on sequencing gels [219,220]. CMCT will modify Us, Ψs, and Gs, but is easily removed from Gs and Us by mild alkaline treatment, leaving CMCT adducts only on Ψ. This bulky adduct blocks reverse transcriptase at sites of pseudouridylation. While these techniques were instrumental for establishing patterns of modification, they were not scalable. Since then, a large number of biochemical, chromatographic, and mass-spectrometric techniques have been developed. Over the past few years, new high-throughput and next-generation sequencing technologies in combination with biochemical approaches have drastically changed the field as they provide unprecedented information on the distribution, regulation, and functional dynamics of the RNA modifications [221,222,223]. These sequencing technologies in the mapping modified RNA or epitranscriptomics’ marks and rely heavily on features such as, pre-enrichment capability (such as antibody enrichment), base-pairing interferences due to modified nucleotides, and bioinformatics tools [222,224,225,226].

Several high-throughput techniques have emerged to specifically analyze both Ψ and 2′Ome. Multiple techniques aim at identifying 2′*O*-me in RNA and have been particularly useful in mapping this modification in rRNA [227]. These include RiboMeth-Seq [215], 2′*O*-Me-Seq [228], RiboOxi-Seq [229], and Nm-Seq [230]. RiboMeth-Seq has become one of the most widely utilized techniques and relies on the fact that 2′-O-methylated sites are more resistant to alkaline hydrolysis by several orders of magnitude when compared to unmethylated site. RNA is randomly hydrolyzed to short RNA fragments and small RNA sequencing libraries are prepared. Reads are aligned to rRNA and the location of 5′ and 3′ ends are determined. These ends reflect sites of hydrolysis and should be randomly distributed. Sites of methylation will be absent when mapped, revealing their location. CMCT modification has also been adapted for a high-throughput approach to map Ψ. Using the original approach, rather than analyzing single sites on sequencing gels, a linker is ligated to the 3′ end of fragmented RNAs, followed by reverse transcription. The Ψs can be mapped by analyzing RT stops after mapping to the genome [231,232,233].

Using these approaches, modification heterogeneity has been demonstrated by several groups and revealed with regards to 2′-O-methylation and pseudouridylation. Using RiboMethSeq in isogenic HCT116 cells that were either p53-positive or p53-negative, the Lafontaine and Motorin labs showed that loss of p53 affected the occupation of 2′-*O*-methylatiom at particular sites [234]. Sites of variability were largely found on the periphery of the ribosome rather than at functional centers. These data were mirrored by earlier work from the Nielsen lab that showed that rRNA from HeLa cells and HCT116 cells were differentially modified [235]. The changes in modification may reflect the changes seen in snoRNAs levels during the development of cancer. However, this was recently extended beyond the pathological setting when it was shown that there were developmentally regulated changes in rRNA methylation during mouse development [236]. In this instance, at least one differentially modified nucleotide occurs as a result of developmental regulation of the Gas5 gene that hosts several snoRNAs. The development of a quantitative mass spectroscopy-based approach to identify modified nucleotides called “stable isotope-labeled ribonucleic acid as an internal standard (SILNAS)” has revealed that, in addition to changes in 2′-*O*-methylation, there are specific sites of pseudouridylation that are differentially modified [3]. Alterations in snoRNA expression seen in disease states may be responsible for some of these changes seen in rRNA modification in cancer. Further work to determine the functional consequences of altered rRNA modification on ribosome activity in these contexts remains to be completed, but it raises many intriguing possibilities.

## Abbreviations/Nomenclature

RNAribonucleic acidbpbase pairntnucleotiderRNAribosomal RNAmRNAmessenger RNAsnoRNPsmall nucleolar ribonucleoproteinsnoRNAsmall nucleolar RNAsnRNAsmall nuclear RNAGguanosineCcytosineAadenosineTthymidineUuridineΨpseudouridine2′Ome2′-*O*-methylRMRPRNA component of mitochondrial RNA processing endonucleaseTERCtelomerase, RNA componentSNU13small nuclear ribonucleoprotein 13NOP58nucleolar protein 58NOP56nucleolar protein 56FBLfibrillarinCRISPRclustered regularly interspaced short palindromic repeatsNHP2non-histone Protein 2NOP10nucleolar Protein 10Gar1glycine/arginine rich-domain containing protein 1DKC1dyskerinCbf5centromere binding factor 5TBF1TTAGGG repeat-binding factor 1Rap1repressor/activator site binding protein 1NME1nuclear mitochondrial processing endoribonucleaseRNaseribonucleaseTGS1trimethylguanosine synthetase 1RPS3ribosomal protein small subunit 3RPL7Aribosomal protein large subunit 7ARPL13Aribosomal protein large subunit 13AGAS5growth arrest specific 5mTORmammalian target of rapamycinHSP90heat shock protein 90R2TPRvb1, Rvb2, Tah1, Pih1 complexRvb1RuvB-like protein 1Rvb2RuvB-like protein 2Tah1TPR repeat-containing protein associated with Hsp90Pih1protein interacting with Hsp90 1RUVBL1RuvB-like AAA ATPase 1RUVBL2RuvB-like AAA ATPase 2RPAP3RNA polymerase II associated protein 3RIH1D1PIH1 domain containing 1snR10snoRNA 10snR30snoRNA 30Sbp1suppressor of PaB mutant 1SSUsmall subunitDNasedeoxyribonucleasehU3-55Khuman U3 component, 55 kDaRrp9ribosomal RNA processing enzyme 9cDNAcomplementary DNACRACcrosslinking and high-throughput analysis of cDNAsSof1suppressor of fibrillarin 1Mpp10M-phase phosphoprotein 10MDamegadaltonrDNAribosomal DNAU3uridine-rich RNA 3ETSexternal transcribed spacerITSinternal transcribed spacerSSvedberg unitsMRPmitochondrial RNA processing enzymeCas9CRISPR-associated protein 9UTRuntranslated regionKRE33/RRA1killer toxin resistant 33/ribosomal RNA cytosine acetyltransferase 1NAT10*N*-Acetyltransferase 10MEFmouse embryonic fibroblastKRASkirsten rat sarcoma viral oncogene homologRASrat sarcomaEMelectron microscopytiRNAtRNA-derived stress-induced RNAtRFtRNA fragmentYB-1Y-box protein 1CMCT*N*-Cyclohexyl-*N*′-(2′morholinoethyl)carbodiimide metho-*p*-toluenesulfonateHCT116human colon cancer 116

## Figures and Tables

**Figure 1 biomolecules-10-00783-f001:**
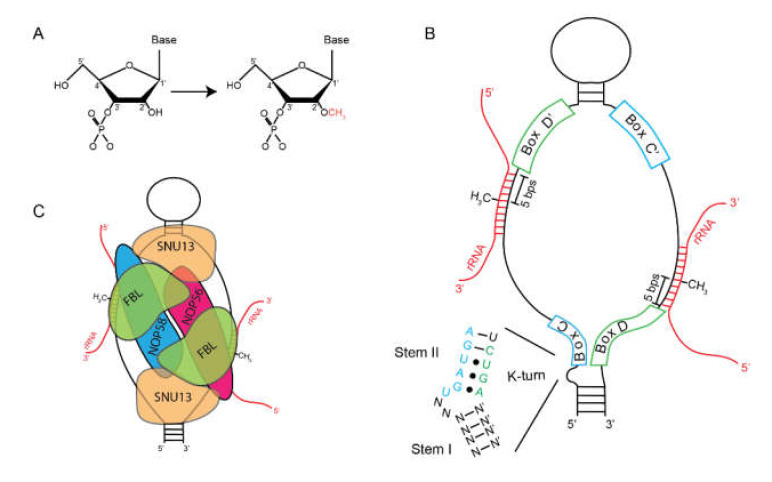
Box C/D snoRNPs. (**A**) Box C/D snoRNPs catalyze the methylation of the 2′ hydroxyl of RNA. This is thought to reduce the hydrophilic nature of the nucleotides and allow rRNA to be buried inside of the ribosome. (**B**) Secondary structure of a typical Box C/D snoRNA indicating the location of Box C/C′ (blue), Box D/D′ (green), and hybridized rRNA (red). Location of methylation is denoted as 5′ bps from Box D/D’. (**C**) Assemblage of protein factors on the snoRNA illustrates that SNU13 binds the K-turns which positions FBL at the site of methylation.

**Figure 2 biomolecules-10-00783-f002:**
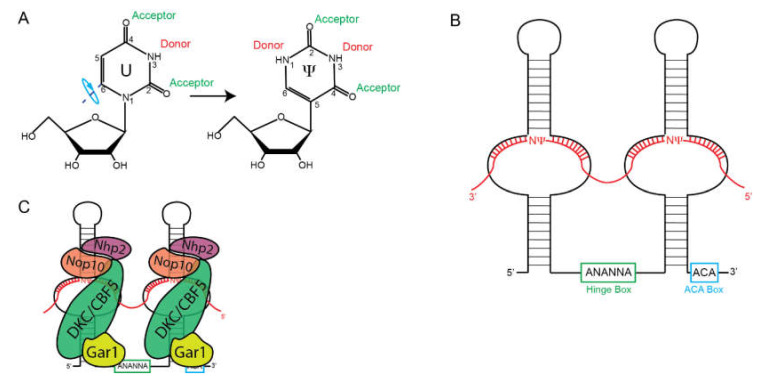
Box H/ACA snoRNPs. (**A**) Isomerization uridine to pseudouridine is catalyzed by Box H/ACA snoRNAs. This generates additional hydrogen bonding capacity that aids in maintaining the ribosome structure. (**B**) Secondary structure of the Box H/ACA snoRNAs, indicating the location of the hinge (H) box (green), ACA box (blue), and hybridized rRNA (red). Pseudouridylation occurs in the “pseudouridylation pocket” at the base of the hairpins. (**C**) Assemblage of protein factors on the snoRNA.

**Figure 3 biomolecules-10-00783-f003:**
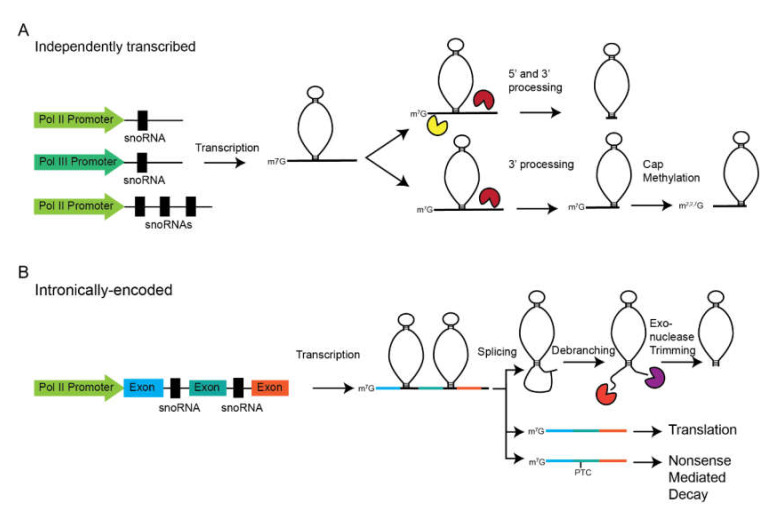
SnoRNA processing pathways. SnoRNAs are either independently transcribed or intronically encoded. (**A**) Independently transcribed snoRNAs exist as mono- or polycistronic units that are transcribed by RNA polymerase II or III. A trimethylguanosine (m^2,2,7^G) cap can be added or the 5′ end can be cleaved off. The 3′ end processing is required for final maturation of both capped and uncapped snoRNAs. (**B**) Intronically encoded snoRNAs are transcribed by RNA polymerase II from the promoters of their host genes. The host gene can encode a functional protein or a noncoding transcript that is often degraded by nonsense-mediated decay due to the presence of a premature termination codon (PTC). Maturation of the snoRNA requires mRNA splicing and debranching of the snoRNA-containing intron followed by exonucleolytic trimming. We presented Box C/D snoRNAs, but Box H/ACA snoRNAs undergo the same maturation pathway.

**Figure 4 biomolecules-10-00783-f004:**
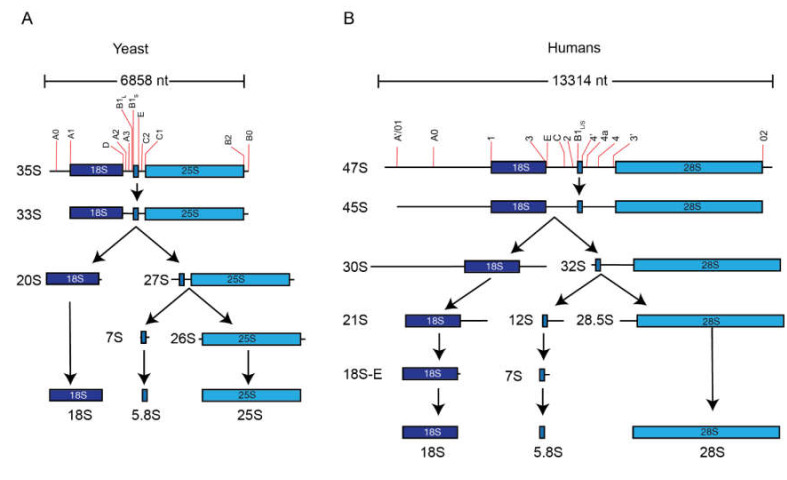
Simplified pathway of rRNA processing. (**A**) Yeast and (**B**) human rRNAs are generated from long polycistronic precursor molecules that must be processed to generate the mature rRNAs. This represents a simplified rRNA processing pathway in each organism, based on [94]. Note that there are minor pathways that are omitted from this schematic.

**Figure 5 biomolecules-10-00783-f005:**
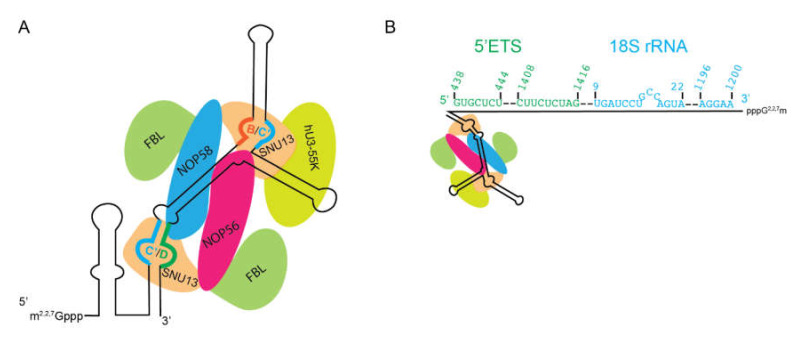
U3 snoRNP. (**A**) Secondary structure of the U3 snoRNA with protein components. Note that SNU13 still binds k-turns, here formed by Box C′ (blue) and Box D (green) and Box C′ (blue) and Box B (Red). The altered structure of the U3 snoRNA results in FBL not being positioned for 2′-O- methylation. (**B**) The 5′ extension forms multiple base-pairs with the pre-rRNA. Two different stretches in the 5′ETS (green) and the 18S rRNA (blue) are bound. Dashed lines indicate looped-out stretches of rRNA. Numbers indicate position in the 47S rRNA (green) or 18S rRNA (blue). Note that this schematic does not fully represent looping out. Secondary structure modified from predictions on snoRNABase [95].

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
