# Peer review of "snoRNPs: Functions in Ribosome Biogenesis"

_biomolecules, 2020, doi:10.3390/biom10050783_

Round 1

Reviewer 1 Report

The review snoRNPs: Functions in Ribosome Biogenesis by Ojha at all is devoted to a very interesting aspect of rRNA modification. It covers the field very good.

I have only some minor comments:

1. After each chapter some "take home message" summarising the main conclusions from described data will be helpful. 

2. Authors should exclude repetitive data from different parts pf the review.

3. Information about snoRNA modification should be included.

4. The comprehensive analysis of the role of rRNA modifications in bacterial ribosome has been recently published: 

Comprehensive Functional Analysis of Escherichia coli Ribosomal RNA Methyltransferases.Pletnev P et al, Front Genet. 2020 Feb 27;11:97. doi: 10.3389/fgene.2020.00097. eCollection 2020.

that should be at least mentioned in the review.

5. English should be checked, sometime it is difficult to read particular sentences.

Author Response

We thank the reviewer for their comments. In response, we have updated the text to remove repetitive sections, particularly those regard use of autoantibodies. We have included a section at the end of each paragraph about "take home messages". We have included the reference to modifications in E. coli and have updated the English for readability. All changes are noted in red.  

Reviewer 2 Report

The review “snoRNPs: Functions in Ribosome Biogenesis” by Sandeep Ojha, Sulochan Malla and Shawn M. Lyons is focussed on the structure and function of snoRNPs in pre-rRNA processing and ribosome assembly.The first part presents an overview of the structure, composition and processing of snoRNAs. Special emphasis is put on methylation and pseudouridylation guides, the Box C/D and H/ACA snoRNPs respectively. The role of snoRNPs in rRNA processing and ribosome function is the main scope of this review. It also includes by a presentation of diseases including congenital disorders and cancer caused by mutations of catalytic components of snoRNPs.

This review presents a simplified view of snoRNP structure/function that often lacks precision. The following points should therefore be improved and several errors should be corrected:

The structure of box C/D snoRNPs is poorly described. Numerous 3D structures are available that highlight the key features of the K-turn (the tandem sheared G-A base pairs and the conserved bulged U) and their importance for snoRNP protein assembly. They should be referenced, so are structures of H/ACA snoRNPs.

Constitutive proteins of box H/ACA snoRNP and their locations should be represented in Figure 2.

snoRNA maturation is coupled to the assembly of core proteins. Although having unrelated structures both box C/D and H/ACA snoRNPs require major chaperone complexes for their assembly, in particular the HSP90/R2TP chaperone-cochaperone system. This could be mentioned.

A schematic representation of the genomic organisation of the snoRNAs and their major processing steps could be included in the review.

Understanding the role of modifications in ribosome function has recently benefited from the emergence of new epitranscriptomic sequencing technologies. Mapping and quantification of RNA modifications at the genome-wide level (Motorin, Helm, Nielsen, Lafontaine,…) have suggested that 2'-O-methylation and pseudouridylation are not all constitutively present on ribosomes. These modifications seem to represent an important source of ribosomal heterogeneity could provide functional specificity to human ribosomes and contribute to the translational control of gene expression. It would be worth highlighting some of these new results.

Bibliographic references are absent in the pre-rRNA processing paragraph and in the legend of Figure 3. This should be corrected.

Mistakes to correct:

Figure 1: the rRNA has two 3’ ends on the Figure 1 B and 1C

Figure 1B: The nature of the canonical K-turn motif outlined Figure 1B should be specified.

Figure 2: An NH group should be present at position 1 of pseudouridine. This should be corrected.

Line 89: Authors write that « uridine is rotated of 180° around C6 ». I guess authors meant around the C6-N3 axis.

Author Response

We thank the reviewer for their comments and have updated our manuscript which we believe has greatly improved it. 

We agree that the original section regarding biogenesis of snoRNAs was abbreviated and we have expanded it. Our original intent was to only provide the basics necessary for a reader to understand so the they would be better equipped to understand the sections regarding ribosome function and processing later. 

We have expanded the section regarding the structure of the K-turn and included several citations on structural analysis of this motif. 

We have included a figure showing the location of proteins in the H/ACA snoRNPs. 

We have included a paragraph describing the R2TP chaperones. 

We have include a figure better describing the genomic context and processing of snoRNAs. 

We have expanded the section regarding new technologies to assay RNA modifications and what this has shown us about ribosome heterogeneity.

We have fixed the 5' and 3' ends of the RNA, added the missing hydrogen, and changed the text from "180o around the C6 axis" to "C6-N3 axis"

Round 2

Reviewer 2 Report

The manuscript “snoRNPs: Functions in Ribosome Biogenesis” by Sandeep Ojha, Sulochan Malla and Shawn M. Lyons has been significantly improved by the expansion of several sections, the addition of a figure and corrections. This is particularly the case for the paragraphs regarding biogenesis of snoRNAs and new technologies to assay RNA modifications. Addition of take home messages at the end of each paragraph also adds clarity of the manuscript. These improvements make the manuscript a comprehensive and informative review.